# Cellular Interplay in COVID-19: Insights from Graph Neural Networks with Multidimensional Edge Features

## Abstract

The COVID-19 has emerged as a global pandemic, posing a significant public health threat with its widespread infection and the potential for severe respiratory complications. Among the various methodologies employed, single-cell omics-based studies have been at the forefront, concentrating on "intra"-cellular properties exhibited by gene expression. However, given its infectious nature, complex biological processes, such as immune responses performed between immune cells, infected cells, etc., necessitate a deeper analysis on "inter"-cellular properties exhibited by cell-cell interaction scores calculated using ligand and receptor expression information. The differences in these interactions in addition to gene expression between severe and non-severe cases could be pivotal in understanding the disease's onset and progression, including mechanism leading to disease severity. Since the structure representing the overall nature of immune response can be implemented by directed graph with cell types as nodes and their interactions as edges, we employed a Graph Neural Network (GNN) model architecture accommodating multi-dimensional edge features, one of the first applications in biological context. In this study, our model incorporates edge features of cell-cell interaction scores, and node features of transcriptional factors and their target genes, which are "intra"-cellular features affected in the downstream by "inter"-cellular features. By leveraging the power of GNNs and the innovative use of multiple edge features, our model offers a groundbreaking perspective on the biological complexity of COVID-19, holding promise for the development of more effective treatments and preventive measures.

## 1 Introduction

The COVID-19 pandemic has had a profound global impact, leading to an unprecedented surge in research to understand the virus and develop potential treatments. Single-cell omics-based studies have emerged as a leading approach in this endeavor, revealing unique cell subtypes and gene markers in COVID-19 patients and providing critical insights into the disease's mechanisms and progression (Van de Sande et al., 2023; Edahiro et al., 2023; Prebensen et al., 2023; Su et al., 2020; Tian et al., 2022; Schulte-Schrepping et al., 2020). However, a comprehensive understanding of COVID-19 requires more than just identifying cellular or genetic markers. Given its infectious nature, the intricate biological processes, such as immune responses, demand a deeper exploration of Cell-Cell interactions (CCI). The variations in these interactions between severe and non-severe cases, or between abnormal and normal cell types, could be key to deciphering the disease's onset and progression.

In our study, we utilized known domain knowledge to construct the graph model (Browaeys et al., 2020; Zhang et al., 2021). Specifically, we integrated node features composed of transcription factors (TFs) and their target genes. The edge features were designed to represent the Ligand-Receptor interactions, which are crucial for understanding cellular communication. Additionally, the TF-Target gene interactions provide insights into the regulatory mechanisms within cells. By combining these features, we aimed to bridge Cell-Cell interactions with the established Ligand-Receptor-Transcription Factor-Target Gene axis.

In addressing this biological complexity, we adopted a novel approach using Graph Neural Networks (GNNs). While GNNs are gaining traction in bioinformatics and single-cell omics, many existing models predominantly utilize single-edge features (Wang et al., 2021; Ma et al., 2023; Efremova & Teichmann, 2020). Our model stands out by being one of the pioneering efforts to incorporate multiple edge features. We meticulously constructed networks for each patient sample, representing each cell type as a node and CCI as edges. This approach, enriched by multiple edge features, offers a more detailed and biologically accurate portrayal of the intricate interactions among different cell types, shedding light on the multifaceted landscape of COVID-19.

We embarked on a comprehensive performance comparison of various GNN architectures. Our exploration spanned from the embedding space of raw node features to the transformed spaces post-convolution and pooling layers. This deep dive enabled us to understand the evolution of data representations at different stages of the GNN models. Furthermore, in our quest to understand performance variations across specific datasets, we delved into the distribution of these datasets. This investigation provided insights into potential reasons for performance discrepancies, emphasizing the importance of dataset characteristics in model outcomes.

Harnessing the capabilities of GNNs, coupled with our innovative use of multiple edge features and thorough performance analysis, our research offers a pioneering perspective on the biological intricacies of COVID-19. This approach not only enhances our understanding of the disease but also paves the way for the development of more effective treatments and preventive strategies.

## 2 METHOD

### 2.1 USED DATASETS

Table 1: Dataset sample distribution

|  |  | | Discovery | | | External | |
|  |  | All | Train | Validation | Test | 1 | 2 |
|---|---|---|---|---|---|---|---|
|  | Healthy | 128 | 77 | 26 | 25 | 11 | 16 |
|  | non-ICU | 325 | 191 | 69 | 65 | 20 | 91 |
| Sample | ICU | 131 | 78 | 28 | 25 | 17 | 52 |
|  | Total | 584 | 346 | 123 | 115 | 48 | 159 |
| Patient |  | 313 | 185 | 64 | 64 | 35 | 97 |

In our study, we utilized single cell RNA sequencing datasets derived from PBMCs, focusing on samples that corresponded to the severity of the patients: Healthy, non-ICU, and ICU. The Discovery dataset comprises samples from 313 patients, totaling 584 samples. This dataset was further segmented into training, validation, and test sets, with 346, 123, and 115 samples respectively. Additionally, we incorporated two external datasets. The first external dataset (External 1) consists of 48 samples from 35 patients, and the second (External 2) includes 159 samples from 97 patients (Schulte-Schrepping et al., 2020; Su et al., 2020). Both external datasets were sourced and preprocessed from Tian et al. (2022). The distribution of samples based on severity across all datasets is detailed in the provided table.

### 2.2 PRE-PROCESSING SINGLE CELL RNA SEQUENCING DATA

We utilized the COVID-19 PBMC single-cell RNA-sequencing preprocessing approach recommended by Tian et al. (2022), but with enhanced quality control standards. Seurat was employed to preprocess each single-cell RNA-seq sample (Hao et al., 2021). Cells not meeting the criteria of nFeature >200 or percent.mt <50 were excluded. We annotated cell types using the PBMC reference single-cell dataset and Seurat's data transfer method (Stuart et al., 2019). Only cells with a mapping, prediction score of level 2 annotation above 0.6 were retained. Additionally, we considered only genes expressed in over 5 cells. Lastly, cells annotated as doublets were removed.

## 2.3 NODE FEATURES

We constructed node features for each cell type based on gene expression using the triMean function from CellChat (Jin et al., 2021). To create node features, we utilized the SpatalkDB (Shao et al., 2022). The SpatalkDB contains information not only about Ligands and Receptors involved in cell-cell interactions but also about Transcription factors and their target genes. From this database, we selected genes annotated as either transcription factors or target genes of transcription factors. Out of the 2,527 genes identified from the database, we chose 1,217 genes that were expressed in the discovery dataset. If a particular gene's expression was not detected in a specific cell type of a given sample, we filled it with a value of 0 to maintain consistent node feature dimensions.

We then performed dimensionality reduction on the raw node feature matrix using scVI (Gayoso et al., 2022). We employed the ModelTuner from the autotune module in scvi-tools for hyperparameter search. Training and tuning were conducted on the discovery train and validation datasets. The search space was set with the number of hidden channels as 64, 128, 256; layer counts as 1, 2, 3; latent dimension counts as 16, 32; learning rates ranging from 1e-4 to 1e-2; gene expression's prior distribution as either zero-inflated negative binomial distribution (ZINB) or negative binomial distribution (NB); and a maximum epoch of 400. The model with 1 layer, 256 hidden channels, 16 latent dimensions, a learning rate of 0.00839, and gene likelihood as ZINB exhibited the lowest loss. Using this model, we reduced the expression of 1,217 genes to 16 dimensions.

## 2.4 EDGE FEATURES

To construct the adjacency matrix and edge feature matrix between nodes, we utilized CellchatDB and the Cellchat function (Jin et al., 2021). Scoring was conducted with min.cells set to 10, and cell-cell interaction scoring was carried out for each sample. All interactions detected in the discovery dataset were employed as edge features. Within the discovery dataset, a total of 313 Ligand-receptor pair level interactions and 107 pathway-level interactions were identified. If any pathway-level interaction between nodes was inferred, the nodes were connected in the adjacency matrix; otherwise, they remained unconnected. The edge feature matrix was constructed using the scores of pathway-level interactions between connected nodes. In instances where a specific node pair (cell type pair) lacked a particular interaction, we filled it with a value of 0 to ensure consistent edge feature dimensions.

## 2.5 GRAPH NEURAL NETWORK MODEL

In our study, we employed several Graph Neural Network (GNN) models, each leveraging unique convolutional mechanisms, especially designed to utilize multi-dimensional edge features. These models, implemented in PyTorch Geometric (Fey & Lenssen, 2019), are briefly described as follows:

- **PNAConv** (Corso et al., 2020): The Principal Neighbourhood Aggregation (PNA) convolutional layer aggregates features from neighboring nodes using a combination of scalers and aggregators. This combination allows for a more flexible and expressive feature aggregation mechanism, capturing complex patterns in the data.

- **GENConv** (Li et al., 2020): The GEN convolutional layer enhances node features by adding transformed edge features to them. This addition operation allows the model to incorporate edge information directly into the node feature space, enriching the representation.

- **NNConv** (Gilmer et al., 2017): The Neural Network convolutional layer uses a neural network to transform edge features to the same dimension as node features. It then performs an element-wise product between the transformed edge features and source node features, allowing for a more intricate interaction between nodes and edges. We'll call this model MPNN afterwards.

- **GATConv** (Veličković et al., 2017): The Graph Attention Network (GAT) convolutional layer computes attention coefficients by considering both source and target node features, as well as the edge features connecting them. These attention coefficients are then used to weigh the contribution of neighboring nodes during feature aggregation.

- **TransformerConv** (Shi et al., 2020): Inspired by the Transformer architecture, this convolutional layer computes attention coefficients by considering transformed node and edge features. Unlike GATConv, TransformerConv incorporates edge features directly in the message-passing step, allowing for a more comprehensive feature aggregation.

By leveraging these convolutional mechanisms, our models effectively utilize multi-dimensional edge features, enhancing their ability to capture intricate patterns and relationships in the graph data.

## 2.6 EXPERIMENTAL SETUP

We split the discovery set into train, validation, and test sets in a 3:1:1 ratio, ensuring class stratification. Since some samples came from the same patient, we assigned samples from the same patient to the same set to prevent potential information leakage. Additionally, we included two external test datasets with available ICU class labels for external validation.

We employed HyperOpt, which is based on bayesian optimization algorithms, to determine the optimal hyperparameters (Bergstra et al., 2013). We experimented with several optimizers, including Adam and SGD (Kingma & Ba, 2014). For SGD, we incorporated a momentum of 0.9 and chose a decay rate between $10^{-5}$ and $10^{-2}$. With Adam, we explored multiple learning rate schedulers, such as cosine annealing warmup restarts, Katsura (2020) (a modification of cosine annealing warm restarts, Loshchilov & Hutter (2016)), ReduceLROnPlateau, and ExponentialLR. The search space for the initial learning rate ranged from $10^{-2}$ to $5 \times 10^{-2}$, and the dropout rate ranged between 0.2 and 0.6. We selected either one or two layers for the number of convolution layers and the number of subsequent linear layers. The hidden channel dimension in the linear layer was chosen from between 8 and 16, while that in the convolution layer ranged from 16 to 32. For some convolution layers, we selected aggregators from options like add, mean, max, min, sum, powermean, and softmax, as accepted by the models. The pooling method, positioned between the convolution and linear layers, was either mean, max, or add. We implemented early stopping based on validation set accuracy with the maximum epoch of 150.

For performance evaluation, we reported accuracy and also provided the weighted F1 score and weighted-averaged AUPRC to assess the model's robustness in handling class imbalances.

## 3 RESULT

### 3.1 GRAPH STATISTICS

Table 2: Graph statistics

|  |  | **Graphs** | **Nodes** | **Edges** |
|---|---|---|---|---|
| Discovery | Total | 584 | 11,978 | 103,997 |
|  | Average per Sample | 1 | 20.51027 | 178.0771 |
| External 1 | Total | 48 | 1,086 | 7,236 |
|  | Average per Sample | 1 | 22.625 | 150.75 |
| External 2 | Total | 159 | 3,622 | 17,622 |
|  | Average per Sample | 1 | 22.77987 | 110.8302 |

In our study, we analyzed graph structures derived from the Discovery, External 1, and External 2 datasets. Table 2 presents a detailed overview of the graph metrics for each dataset For all datasets, each sample was represented by a single graph. The node features and edge features remained consistent across datasets, with 16 node features and 107 edge features. The mean number of nodes and edges per sample varied slightly across datasets, with the Discovery dataset having the highest average number of edges per sample.

Table 3: Model performance metrics

| Model | Discovery | External 1 | External 2 |
|---|---|---|---|
| **Accuracy (%)** | | | |
| PNA | 80.00 ± 3.61 | 57.08 ± 3.86 | 59.25 ± 2.19 |
| GEN | **80.35 ± 2.61** | **73.33 ± 5.17** | **58.24 ± 4.82** |
| GAT | 76.87 ± 3.55 | 59.58 ± 10.42 | 50.44 ± 9.32 |
| Trans | 78.78 ± 1.79 | 64.17 ± 5.80 | 57.61 ± 2.34 |
| MPNN | 79.30 ± 3.50 | 69.58 ± 6.67 | 58.49 ± 4.13 |
| **Weighted F1 Score (%)** | | | |
| PNA | 80.19 ± 3.51 | 55.08 ± 3.90 | 49.08 ± 4.38 |
| GEN | 80.43 ± 2.52 | **71.78 ± 6.92** | 47.02 ± 4.18 |
| GAT | 77.08 ± 3.42 | 55.61 ± 14.36 | 48.84 ± 7.38 |
| Trans | 78.92 ± 1.66 | 59.61 ± 9.46 | 47.03 ± 3.56 |
| MPNN | 79.47 ± 3.21 | 67.15 ± 9.72 | 47.53 ± 3.77 |
| **Weighted-averaged AUPRC (%)** | | | |
| PNA | 85.33 ± 2.43 | 71.38 ± 3.95 | 58.87 ± 1.93 |
| GEN | 84.05 ± 0.33 | **83.87 ± 1.20** | 60.68 ± 3.52 |
| GAT | 83.84 ± 1.73 | 75.41 ± 11.12 | 58.59 ± 3.98 |
| Trans | 81.02 ± 1.30 | 77.59 ± 4.83 | 55.81 ± 4.20 |
| MPNN | 84.76 ± 1.63 | 83.03 ± 2.71 | 58.03 ± 1.47 |

*Note: Values are presented in mean ± s.d. obtained from 5 runs with 5 random seeds*

## 3.2 MODEL PERFORMANCE

In our study, we evaluated the performance of various graph-based classifiers, including PNA, GEN, GAT, Trans, and MPNN, across three datasets: Discovery, External 1, and External 2. The classifiers were assessed based on three metrics: Accuracy (Acc), Weighted F1 score, and Weighted-averaged AUPRC. (See Table 3)

For the Discovery dataset, all classifiers demonstrated comparable performance, with GEN achieving the highest accuracy of 80.35±2.61%. In the context of the External 1 dataset, GEN outperformed the other models by a significant margin, achieving an accuracy of 73.33±5.17%, which was notably higher than the second-best performing model, MPNN, which achieved 69.58±6.67%. For the External 2 dataset, PNA and GEN were closely matched in terms of accuracy, with GEN slightly underperforming at 58.24±4.82%.

When considering the Weighted F1 score, GEN consistently ranked among the top performers across all datasets. Particularly in the External 1 dataset, GEN achieved a score of 71.78±6.92%, surpassing the other models by a significant margin.

In terms of the Weighted-averaged AUPRC, GEN showcased superior performance, especially in the External 1 dataset, where it achieved a score of 83.87±1.20%, the highest among all classifiers.

Given the comprehensive evaluation, we selected the GEN classifier for our study due to its consistent top-tier performance across all datasets and metrics. Its robustness and adaptability to different datasets, as evidenced by the results, made it the most suitable choice for our research objectives.

## 3.3 VISUALIZATION OF LEARNED REPRESENTATIONS IN GNN

We visualized the learned representations based on UMAP results as dimension reduction, message passing, pooling is done along the model (McInnes et al., 2018). We chose the model using GEN-Conv as the convolution layer to show the representations since it showed fairly good performance overall. Out of 5 runs from 5 random seeds, we selected the one with fairly performing in all datasets based on accuracy. In Figure 1(a), raw embeddings are constructed from gene expression values possibly being affected by interactions between immune cells, represented by edge features. It was able

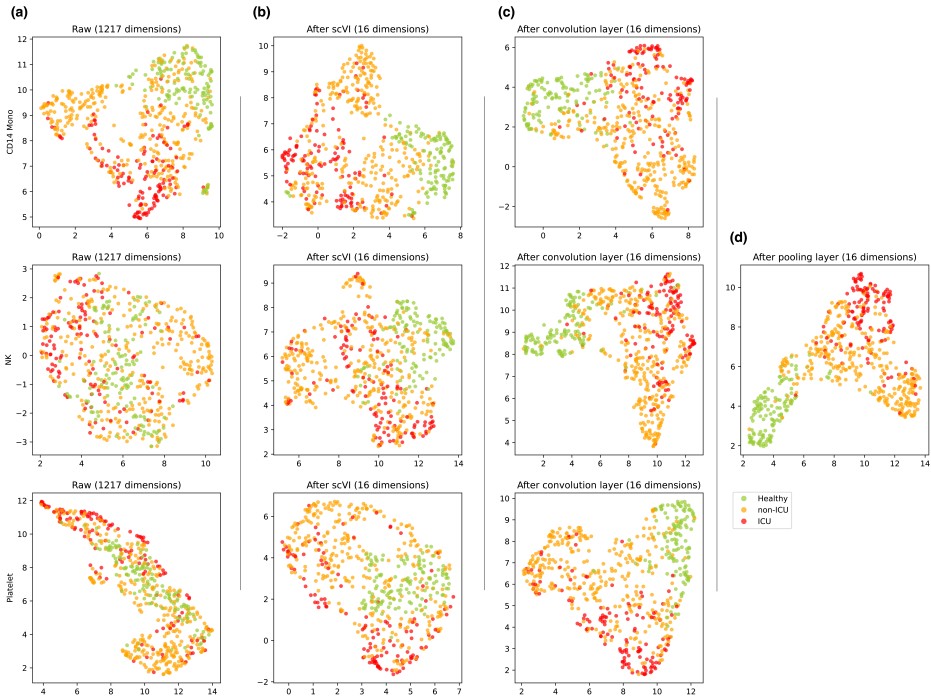

Figure 1: UMAP results of learned node-level / sample-level representations from each layer of our model. **(a)** Raw gene expressions of 3 major cell types, CD14 Mono, NK, Platelet with 1217 dimensions. **(b)** After scVI (dimension reduction) with 16 dimensions. **(c)** After convolution layers with 16 dimensions **(d)** After pooling layer with 16 dimensions **(a)**-**(c)** is node-level embeddings from each cell type, and **(d)** is sample-level embeddings.

to identify that some cell types (e.g. CD14 Monocyte) were readily posing difference between ICU classes, opposed to other cell types (e.g. NK, Platelet), randomly distributed in embedding space. As the dimension reduction is done through scVI, the overall distribution of samples seemed to be preserved as shown in Figure 1(b). Following the effect of message passing in convolution layers, ICU classes were starting to gather together in all cell types, showing more cohesion after pooling node-level representations to sample-level representations. (Figure 1(c, d))

### 3.4 AGE-RELATED VARIATIONS AND THEIR IMPACT ON MODEL GENERALIZATION

External 2 demonstrated a lower performance compared to the Discovery and External 1 datasets. When visualizing the embedding space, the samples from Discovery and External 1 clustered closely, while those from External 2 grouped distinctly (See Figure 2). All datasets provided patient age information, a factor known to correlate strongly with COVID-19 severity (Liu et al., 2020; Molani et al., 2022; Zimmermann & Curtis, 2022). To understand the differences in these datasets, we examined the age distribution within each ICU class. The Wilcoxon test revealed significant age differences between the ICU and non-ICU groups in the Discovery dataset (p-value = 2.6e-05, See Figure 3). Interestingly, the non-ICU group in External 2 had a higher age range than its counterpart in the Discovery dataset, with an average age difference of about 10 years (p-value = 1.2e-06, See Figure 4, Table 4). No such age difference was observed within the ICU class across all datasets. The composition of the non-ICU class in External 2 differed from that in the Discovery dataset. This difference, combined with the age variations, likely introduced biases not present in the Discovery dataset. Such imbalances and variations in patient characteristics might have hindered the model's ability to generalize effectively across datasets, leading to the observed performance decline.

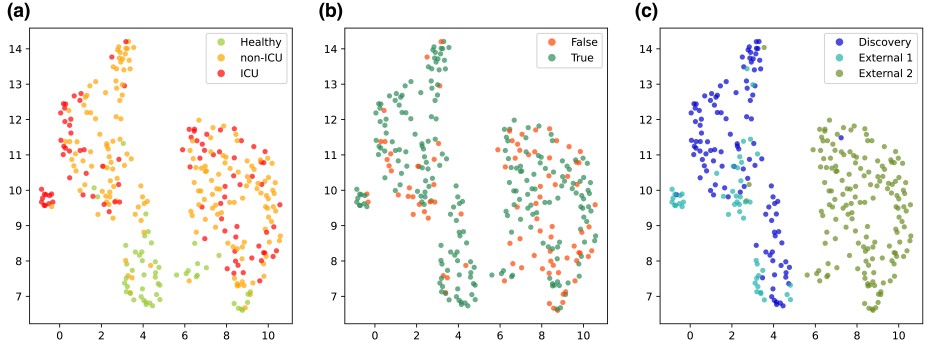

Figure 2: UMAP results of learned latent embeddings after pooling layer of all test datasets. Representations colored by **(a)** Ground truth class label **(b)** Incorrect or correct predictions presented in boolean values **(c)** Datasets

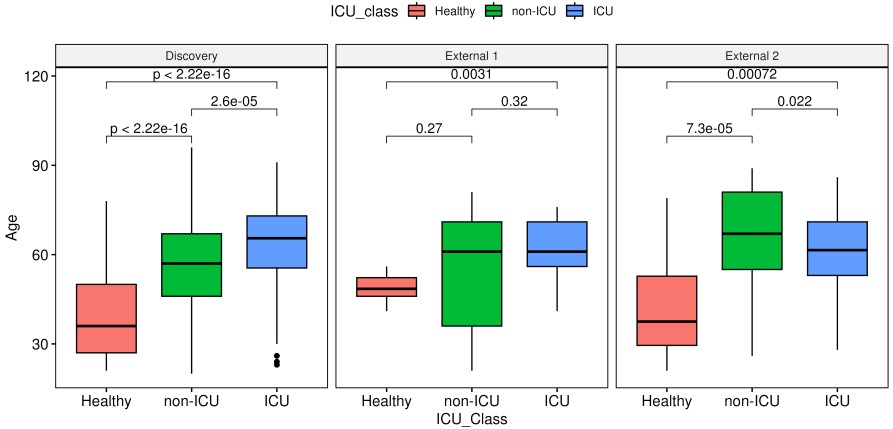

Figure 3: Boxplot showing the age distribution for each class

## 4 DISCUSSION

In this study, we proposed a novel graph structure derived from single-cell transcriptomes, grounded in biological domain knowledge. Recognizing that cells can have multiple simultaneous interactions rather than a singular one, we integrated this complexity into our graph representation. Furthermore, instead of relying on statistical modeling for node feature selection (e.g., genes that statistically differ between classes or highly variable genes within datasets), we opted for a domain knowledge-driven approach, focusing on Transcription Factors (TFs) and their target genes.

We explored various Graph Neural Network (GNN) architectures capable of learning from our proposed graph. These models were then evaluated for their ability to classify sample states (ICU class) using both our discovery dataset and external datasets.

However, there were challenges. The discovery dataset, while comprehensive, might not have had a sufficient number of data points or a balanced composition. This limitation became evident when we observed a significant performance drop in the independently collected External 2 dataset. It underscores the need for more extensive and diverse data collection before building robust models.

Given that our graph structure is rooted in domain knowledge, we anticipate that applying explainable AI techniques to our constructed models will facilitate biological interpretation. By leveraging explainable AI, we aim to holistically explore the Ligand-receptor-TF-target gene axis involved in

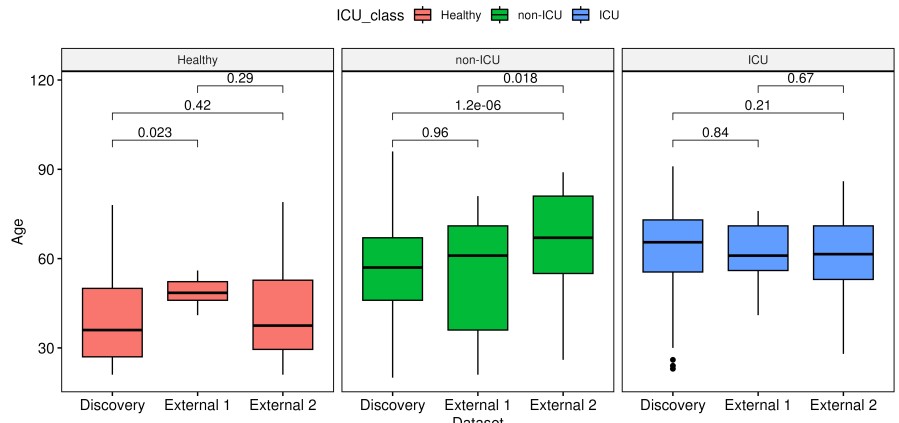

Figure 4: Boxplot showing the age distribution for each dataset

Table 4: Average age distribution across different classes in datasets

|  | **Healthy** | **Non-ICU** | **ICU** |
|---|---|---|---|
| **Discovery** | 39.125 (36.0) | 55.797 (57.0) | 62.564 (65.5) |
| **External 1** | 49.125 (48.5) | 53.632 (61.0) | 62.471 (61.0) |
| **External 2** | 42.571 (37.5) | 65.527 (67.0) | 59.731 (61.5) |

*Note: Values in parentheses represent the median*

COVID-19 severity. This approach holds promise for unveiling intricate biological insights, potentially leading to better understanding and interventions for the disease.

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
