# OpenReview forum: "Cellular Interplay in COVID-19: Insights from Graph Neural Networks with Multidimensional Edge Features"
_ICLR.cc/2024/Conference — Submitted to ICLR 2024_

### Official Review · Reviewer_Fa7B · 2023-10-27

**Soundness:** 2 fair
**Presentation:** 2 fair
**Contribution:** 1 poor
**Rating:** 3
**Confidence:** 4

**Summary:**

Here the authors propose a new method for constructing graphs to represent cell type <-> cell type relations in scRNA-seq data for use with GNN architectures. The authors then apply their method to study data from COVID-19 patients.

**Strengths:**

* **Motivation**: The problem considered by the authors - i.e., gaining insights on COVID-19 from single-cell data - is clearly significant.
* **Novelty**: The authors' method is, to my knowledge, novel.

**Weaknesses:**

Based on my reading of the manuscript, significant changes are needed before this work can be considered ready for publication at a top tier conference. In short: the experiments in the paper are far too limited and the biological insights provided are minimal. Details on my major concerns below:

* **No new biological insights:** In the abstract the authors claim that their model "offers a groundbreaking perspective on the biological complexity of COVID-19". However, this (substantial) claim does not appear to be supported by the experiments presented in the manuscript. While it's reassuring that the proposed graph construction method leads to decent classification of disease severity (Table 3) using standard GNN architectures, this does not provide any new insights on COVID-19. Similar comments apply to the visualizations presented in Figures 1 and 2.
* **No comparisons with GNN baselines:** The only performance comparisons presented by the author consist of different GNN architectures applied with the authors' proposed graph construction technique. Thus, it is impossible to understand how the authors' proposed method performs relative to previously GNN models for single cell data (such as those from Wang et al. 2021, Ma et al. 2023, or Efremova & Teichmann, 2020 cited by the authors.
* **No comparisons with non-GNN baselines:** Beyond the lack of comparison with GNN baselines, it's not obvious to me that GNNs are the right tool for this problem, as opposed to e.g. well-established tools like Seurat/scanpy/scVI/etc. Thus, I believe it's critical that the authors also compared their work with standard non-GNN based workflows. The authors provide some visualizations of the representations learned by their method versus scVI in Figure 2, but qualitatively I don't see any major differences between the methods (i.e., healthy tend to separate from non-ICU/ICU, and there's some mixing between non-ICU and ICU). As a result, without additional results it's impossible to conclude that the authors' method provides any advantages over previous work.
* **Limited discussion/comparison to previous work:** The authors are not the first to apply GNNs to single-cell transcriptome data from COVID-19 patients (see e.g. [1]). However, the authors do not include any discussion of previous similar work in this area, and thus it is difficult to assess how the proposed work relates to previous works in the literature.
* **Limited details on the graph-building technique:** The authors' proposed technique for building the cell type <-> cell type interaction graph relies heavily on previously proposed tools (e.g. SpatalkDB, CellchatDB). However, little information about these tools is provided to the reader and thus it's difficult to understand their exact function. I would highly recommend that the authors remove the extensive discussions of hyperparameter settings (which can be relegated to the supplement) and instead provide more information on the tools used to build their graphs.

[1]: Sehanobish, Ravindra, and van Dijk, "Gaining Insight into SARS-CoV-2 Infection and COVID-19 Severity Using Self-supervised Edge Features and Graph Neural Networks". (2020)

**Questions:**

* How do the authors envision their method providing new biological insights?
* Why were no baseline methods considered in the manuscript?
* How does the proposed method relate to other GNN methods use for analyzing COVID-19 data?

---

### Official Review · Reviewer_MUv9 · 2023-10-30

**Soundness:** 2 fair
**Presentation:** 1 poor
**Contribution:** 1 poor
**Rating:** 3
**Confidence:** 4

**Summary:**

Using scRNAseq data from PBMCs in patients with differing severity of COVID, the authors study the problem of correctly classifying this severity using known information about both cell types and cell-cell interactions.

**Strengths:**

- Incorporates known biology
- Trains five different kinds of GNNs
- Labels each edge with scores from 107 pathway-level interactions

**Weaknesses:**

- The authors claim an "innovative" and "pioneering" use of "meticulously constructed" multidimensional edge features, but multidimensional edge features in GNNs have been around for a long time. Perhaps they mean for this particular problem of classifying COVID severity, but that is hardly a novel contribution.
- The authors claim that these multidimensional edge features make these models very performant, but do not include a single study with a one-dimensional edge features to validate this claim.
- In Table 3, standard deviations are typically large and therefore the best performing models are typically not significantly better.
- The distribution of severity classes in the UMAPs of Figure 1 do not look visually different from each, which weaken the authors' claims that the MPNN is creating greater cohesion after its conv or pooling layers.
- The age-related analysis and its resulting conjecture about model performance on External 2 is weak - there could be many factors contributing to these differences. It would be better to these the hypothesis by, say, resampling External 2 to have a similar age distribution then training.

**Questions:**

- Why not train on Discovery and test on External 1 or 2?
- Why not perform an ablation study on the models to better understand the mechanisms creating differences in performance?
- Why not compare with a simple baseline model?

---

### Official Review · Reviewer_CTJa · 2023-11-02

**Soundness:** 2 fair
**Presentation:** 2 fair
**Contribution:** 1 poor
**Rating:** 1
**Confidence:** 4

**Summary:**

In this paper, the authors employ GNN (graph neural network) for the study of single-cell RNA sequencing data from COVID-19 patient samples.
By defining node features based on transcription factors (TFs) and their target genes and designing edge features to represent various cell-cell interactions, several GNNs were implemented using PyTorch to build classifiers that can discriminate different patient states (i.e., healthy, non-ICU, ICU).
The classification performance of the constructed GNN classifiers is assessed based on the available clinical data and the learned representations are visualized using UMAP and further investigated to gain further insights.

**Strengths:**

This manuscript investigates the potential advantages of using GNNs to incorporate cell-cell interactions for analyzing COVID-19 single-cell RNA sequencing data.
The models were applied to the patient state classification problem (healthy, non-ICU, ICU) and evaluated based on clinical data.
The analysis results presented in this study may provide some useful insights for other researchers who investigate similar bioinformatics & medical informatics problems.

**Weaknesses:**

There are several major concerns about the current manuscript.

While the paper makes bold claims about the novelty and significance of the current study (e.g., "innovative use of multi-edge features", "offers a groundbreaking perspective", "stands out ... one of the pioneering efforts", etc.) there is very limited technical innovation (if any) reported in the current paper nor does it present biologically & clinically significant novel insights based on their COVID-19 data analysis.

The GNNs in this study are simply implemented utilizing existing models in PyTorch, without any additional novel methodological contributions.

The use of multiple edge features are claimed to be "pioneering" and "innovative", but similar efforts have been made in relevant fields (e.g., bioinformatics and computational biology) for many years.
For example, many relevant efforts are reported in a recent review paper by Zitnik et al. entitled "Current and future directions in network biology" (arXiv:2309.08478), which extensively discuss how GNNs (and other network models) are used for in "Network Biology" research and how meaningful representations are learned by incorporating a wide variety of node and edge features.

The paper also claims that "our model offers a groundbreaking perspective on the biological complexity of COVID-19" and that the proposed approach "not only enhances our understanding" but also "pave the way for the development of more effective treatments and preventive strategies" for COVID-19.
Unfortunately, the evaluation results, analysis, and discussions present in this paper do not provide any novel insights that are biologically/clinically significant.

**Questions:**

There are critical concerns regarding the main novel contributions made in this work.

1. Does this study make any significant and novel methodological contributions?
If so, what are these contributions and why are they significant?

2. What novel (and concrete) insights can be drawn based on the results reported in this paper?
Why are they biologically/clinically significant?

These key issues need to be clarified by the authors and need to be clearly stated upfront (i.e., in the abstract and the introduction).

---

### Official Review · Reviewer_vxYD · 2023-11-10

**Soundness:** 2 fair
**Presentation:** 2 fair
**Contribution:** 2 fair
**Rating:** 3
**Confidence:** 4

**Summary:**

The study aims to analyze cell-cell interactions (CCI) using Graph Neural Networks (GNN), particularly ligand-receptor interactions and gene expression. The GNNs are trained on supervised problems: predict patient features using a cell-type by cell-type graphs. The authors apply this method to a COVID-19 dataset.

Claimed contributions:
- Employing GNNs to model multi-dimensional edge features for analyzing CCIs in COVID-19.
- Integrating node features of transcription factors and target genes with edge features of cell-cell interactions.
- Conducting comprehensive performance comparisons of various GNN architectures to understand data representation evolution and performance variations across datasets.

**Strengths:**

- benchmark papers for GNN in this use case are timely
- the proposed featurization strategy could be useful for cell-to-cell network analysis but more info needed (see questions)

**Weaknesses:**

- no "related method" section
- clarity on contributions and methodological details (see questions)

**Questions:**

- Explicit Statement of Contributions: Please provide a more detailed explanation of your specific contributions. It is currently challenging to discern whether the novelty lies in the featurization process, the application of graph architectures, or a new benchmarking of existing architectures. A clearer delineation of what is novel in your work will greatly enhance the reader's understanding and appreciation of your contributions. Also, considering the venue of this submission, it would be appropriate to emphasize the methodological contributions of your work. The title and abstract currently highlight the biological insights into COVID-19, which may not align perfectly with the venue's focus. If the primary contribution is methodological, consider reshaping the paper to foreground these innovations, using the COVID-19 application as a demonstrative example.

- Inclusion of a Related Methods Section: A section discussing related methods and situating your work within the state-of-the-art would be highly beneficial. This would help readers understand how your approach compares to and potentially advances beyond current methodologies in this field.

- Detailed Description of Featurization Process: The featurization process for both edge and node features is central to your methodology but is not thoroughly explained. A more detailed description, or at least a high-level overview with additional details provided in a supplementary section, would be valuable. Including a figure to illustrate this process would greatly aid in comprehension.

- Clarification of Machine Learning Task (presentation / minor): The manuscript would benefit from a clear description of the machine learning task being addressed. It appears to be a supervised machine learning classification problem, but this is not explicitly stated. Clarifying the nature of the task will help readers understand the context and objectives of your study.

These suggestions are intended to enhance the clarity and impact of your manuscript. I am willing to revise my score if the authors can clarify these points.

---

### Meta-Review · Area_Chair_xV9v · 2023-12-10

**Metareview:**

The paper presents a method for analyzing cell-cell interactions (CCI) in COVID-19 using Graph Neural Networks (GNNs), focusing on ligand-receptor interactions and gene expression. The GNNs are trained to predict patient features using cell-type by cell-type graphs, with node features based on transcription factors and target genes, and edge features representing cell-cell interactions. The authors apply this method to a COVID-19 dataset and conduct performance comparisons across various GNN architectures.

Reviewer vxYD raised concerns about the clarity of the paper's contributions and methodological details. They suggested the need for a clearer statement of contributions, inclusion of a related methods section, a detailed description of the featurization process, and clarification of the machine learning task. Reviewer CTJa criticized the paper for overstatement of novelty and significance, lack of significant technical innovation, and absence of novel insights from the COVID-19 data analysis. They questioned the novelty of the methodological contributions and the biological/clinical significance of the insights. Reviewer MUv9 pointed out that the use of multidimensional edge features is not novel and that the paper lacks comparative studies with simpler models. They also noted large standard deviations in the results and questioned the biological insights provided. Reviewer Fa7B expressed concerns about the lack of new biological insights, absence of comparisons with GNN and non-GNN baselines, limited discussion of previous work, and insufficient details on the graph-building technique. They questioned the method's ability to provide new biological insights.

All reviewers agree that while the topic is timely and the approach is potentially interesting, the paper falls short in several key areas as mentioned above. Given these concerns, the consensus among the reviewers is to reject the paper. However, the topic's relevance and the potential of the approach suggest that significant revisions addressing these concerns could make the work suitable for publication in the future.

**Justification For Why Not Higher Score:**

Reviewer vxYD raised concerns about the clarity of the paper's contributions and methodological details. They suggested the need for a clearer statement of contributions, inclusion of a related methods section, a detailed description of the featurization process, and clarification of the machine learning task. Reviewer CTJa criticized the paper for overstatement of novelty and significance, lack of significant technical innovation, and absence of novel insights from the COVID-19 data analysis. They questioned the novelty of the methodological contributions and the biological/clinical significance of the insights. Reviewer MUv9 pointed out that the use of multidimensional edge features is not novel and that the paper lacks comparative studies with simpler models. They also noted large standard deviations in the results and questioned the biological insights provided. Reviewer Fa7B expressed concerns about the lack of new biological insights, absence of comparisons with GNN and non-GNN baselines, limited discussion of previous work, and insufficient details on the graph-building technique. They questioned the method's ability to provide new biological insights. All reviewers agree that while the topic is timely and the approach is potentially interesting, the paper falls short in several key areas as mentioned above.

**Justification For Why Not Lower Score:**

N/A

---

### Decision · Program_Chairs · 2024-01-16

Reject